# West Nile Disease Symptoms and Comorbidities: A Systematic Review and Analysis of Cases

**DOI:** 10.3390/tropicalmed7090236

**Published:** 2022-09-08

**Authors:** Maria Bampali, Konstantinos Konstantinidis, Emmanouil E. Kellis, Theodoti Pouni, Ioannis Mitroulis, Christine Kottaridi, Alexander G. Mathioudakis, Apostolos Beloukas, Ioannis Karakasiliotis

**Affiliations:** 1Laboratory of Biology, Department of Medicine, Democritus University of Thrace, 68100 Alexandroupolis, Greece; 2First Department of Internal Medicine, Democritus University of Thrace, University General Hospital of Alexandroupolis, 68100 Alexandroupolis, Greece; 3Department of Genetics, Development and Molecular Biology, School of Biology, Aristotle University of Thessaloniki, 54124 Thessaloniki, Greece; 4Division of Infection, Immunity and Respiratory Medicine, School of Biological Sciences, The University of Manchester, Manchester Academic Health Science Centre, Manchester M23 9LT, UK; 5The North West Lung Centre, Wythenshawe Hospital, Manchester University NHS Foundation Trust, Manchester M23 9LT, UK; 6Molecular Microbiology & Immunology Lab, Department of Biomedical Sciences, University of West Attica, 12243 Athens, Greece; 7National AIDS Reference Centre of Southern Greece, Department of Public Health Policy, University of West Attica, 11521 Athens, Greece

**Keywords:** West Nile virus, encephalitis, meningitis

## Abstract

West Nile virus (WNV) is a mosquito-borne flavivirus that has emerged as a major cause of viral encephalitis and meningitis, rarely leading to death. Several risk factors have been discussed in the past concerning the severity of the disease, while few reports have focused on precipitating conditions that determine of WNV-related death. Studies on cohorts of patients suffering of West Nile disease (WND) usually encompass low numbers of deceased patients as a result of the rarity of the event. In this systematic review and critical analysis of 428 published case studies and case series, we sought to evaluate and highlight critical parameters of WND-related death. We summarized the symptoms, comorbidities, and treatment strategies related to WND in all published cases of patients that included clinical features. Symptoms such as altered mental status and renal problems presented increased incidence among deceased patients, while these patients presented increased cerebrospinal fluid (CSF) glucose. Our analysis also highlights underestimated comorbidities such as pulmonary disease to act as precipitating conditions in WND, as they were significantly increased amongst deceased patients. CSF glucose and the role of pulmonary diseases need to be revaluated either retrospectively or prospectively in WND patient cohorts, as they may be linked to increased mortality risk.

## 1. Introduction

West Nile virus (WNV) is a widely spread mosquito-borne flavivirus that poses a major health burden to areas where it is endemic [1]. It belongs to the *Flavivirus* genus (*Flaviviridae* family) along with other medically important pathogens, such as dengue, Zika, yellow fever, Japanese encephalitis, tickborne encephalitis, Usutu, Saint Louis encephalitis, and Powassan virus [2]. These viruses have been implicated to a number of severe clinical conditions, such as neurological complications and congenital malformations [3]. Some of them can also cause viscerotropic and hemorrhagic syndrome [2]. A better understanding of the pathogenicity and the reasons behind the manifestation of different clinical symptoms will help treat infected people more effectively, reduce the risk of an adverse outcome, and prevent viral spread.

WNV is one of the most common causes of encephalitis in the world. Most of the people infected with the virus remain asymptomatic (80%) and are not reported [4]. A few infections result in mild flu-like symptoms such as fever, headache, myalgia, and fatigue (20%), a syndrome usually referred to as West Nile fever (WNF) [5]. WNF is the non-neuroinvasive form of the disease and normally lasts about 3 to 6 days after the first onset of the symptoms. Gastrointestinal complications may also be present during this period, including nausea, vomiting, and diarrhea, but this condition is usually self-limited and mild [6]. West Nile neuroinvasive disease (WNND) occurs in less than 1% of the cases when the virus invades the central nervous system. It usually manifests as encephalitis, meningitis, or poliomyelitis-like acute flaccid paralysis [7]. While people of all ages are vulnerable to infection with the virus, the risk of developing severe neurological symptoms increase with age (over 60 years, as well as underlying medical conditions, such as cancer and diabetes [7]. WNV encephalitis is the most common manifestation among WNND patients (50–71%). It usually involves muscular weakness and dysfunction of the peripheral nervous system. WNV meningitis (occurs in 15–35% of the WNND cases) has been associated with an increase of white blood cells in the patient’s cerebrospinal fluid, fever, headache, and stiff neck. WNV poliomyelitis is less common (3–19%) and refers to an acute flaccid paralysis syndrome, which is characterized by the acute onset of limb weakness or paralysis without the common symptoms of an infection, such as fever or headache [4]. Usually, most of the symptoms of WNV infection do not persist for more than a year. However, it is not uncommon for some patients, especially those recovering from the neuroinvasive disease, to experience long-term physical, cognitive, as well as functional complications such as weakness, fatigue, memory loss, depression, and difficulty concentrating. Generally, WNF and WNV meningitis cases have a good prognosis, while WNV encephalitis patients have a higher risk for an adverse outcome, and 10–30% of them may die [7].

Symptoms and comorbidities of WNV infection have been assessed mainly within certain local cohorts or case studies, while prognosis of the disease outcome is still obscure. In order to summarize the findings of current bibliography on symptoms and comorbidities in recovered and deceased patients infected with WNV, we conducted a systematic review using publications indexed in *PubMed*/*Medline* and *EMBASE*.

## 2. Materials and Methods

For conducting and reporting this systematic review, we followed standard methodology recommended by the Cochrane Collaboration [8] and the PRISMA Statement [9], respectively. Eligible studies comprised case reports or case series describing the clinical presentation and/or outcomes of patients with a microbiologically confirmed WNV infection. For this systematic review, we considered relevant data on the clinical and laboratory characteristics, the outcomes, and factors associated with the outcomes of patients with WNV infection.

*PubMed*/*Medline* and *EMBASE* were searched using a structured search strategy that included controlled vocabulary and free search terms aiming to identify studies of relevant design evaluating the presence of WNV in humans (Appendix A). Detailed search strategies are provided in the online appendix. Four investigators (Maria Bampali, Konstantinos Konstantinidis, Manolis Kellis, and Theodoti Pouni) independently screened for eligibility all studies yielded by the searches at a title/abstract level, followed by full-text assessment of all potentially eligible studies. Two investigators independently extracted relevant data of all participants in a structured form that was prospectively designed and piloted among all investigators. Disagreement in any step of the process was resolved by discussion and adjudication by the senior author.

As the case series encompassed small numbers of patients, they were restrictive in terms of formal meta-analysis of the data. Thus, the present report summarizes the findings of all the reports presented to date. Clinical and laboratory characteristics and the clinical outcomes of patients are presented narratively. Differences in the characteristics between recovered, and deceased patients with any WNV-related disease entity are noted.

## 3. Results

For the systematic review of symptoms and comorbidities of WNV infection, we conducted a systematic search of reports on WNV cases using the Rayyan platform [10] (https://www.rayyan.ai, accessed on 17 May 2021, search was conducted 17/05/2021). The search returned 1813 articles, of which 442 (24.4%) were included after assessing whether the article included pathological description of WNV cases. Eighty-one articles (4.5%) were presented as conflicts between the blind collaborators, and the remaining 1290 articles (71.1%) were excluded as irrelevant to disease pathology (Figure 1). The 81 conflicting articles were all excluded after screening. Then, 442 articles were deduped, and the remaining 428 articles (Appendix A) were used to populate a binary table according to the reported symptoms or comorbidities of the reported cases (Appendix A).

The cases were divided between those patients that survived after WNV infection and those who deceased. Statistical analysis on the occurrence rate of certain symptoms and the comorbidities that have been linked in the past with poor outcome revealed similarities and differences between the two groups.

### 3.1. Symptoms and Cellular and Biochemical Measurements

The patients that recovered versus those that deceased showed similar rates of occurrence for the majority of symptoms related to WNV infection. However, patients that deceased presented fever, which is the most common symptom following WNV infection, at a higher rate (~1.2 fold higher [recovered 75.73%/deceased 87.63%], *p* = 0.01195) (Figure 2, Appendix A). The altered mental status, often presented as confusion or delirium, was also significantly more apparent (~1.4 fold higher [recovered 39.77%/deceased 55.67%], *p* = 0.00804) in the same group as compared to the patients that survived. Patients that succumbed more often experienced anorexia and weight loss (~3 fold higher [recovered 3.80%/deceased 11.34%], *p* = 0.00387) and kidney and urine problems (~2.7 fold higher [recovered 2.63%/deceased 7.22%], *p* = 0.03348) (Figure 2, Appendix A). On the other hand, headache (~1.7 fold higher [recovered 38.60%/deceased 22.68%], *p* = 0.00367), walking and balance problems (~2.8 fold higher [recovered 11.70%/deceased 4.12%], *p* = 0.02840), and rash (~2.8 fold higher [recovered 14.62%/deceased 5.15%], *p* = 0.01287) were more prominent in patients that recovered after WNV infection (Figure 2, Appendix A). CSF measurements were readily available in reports as compared to blood measurements for glucose, protein, and white and red blood cell abundance. Amongst CSF measurements, glucose levels in the CSF were significantly higher (*p* = 0.0003) in patients that deceased (Figure 3, Appendix A).

### 3.2. Comorbidites

Fatal outcome in West Nile disease has been reported to be associated with certain comorbidities. The patients that deceased more frequently presented comorbidities such as cancer (~2.4 fold increase [recovered 11.99%/deceased 28.87%], *p* < 0.0001) and transplantation or transfusion (~2.7 fold increase [recovered 12.57%/deceased 34.02%], *p* < 0.0001) (Figure 4, Appendix A). Interestingly, although at low rates, the cooccurrence of pulmonary infection was significantly higher in patients that deceased (~3.5 fold increase [recovered 1.46%/deceased 5.15%], *p* = 0.03147) (Figure 4, Appendix A). On the other hand, comorbidities usually associated with poor prognosis of West Nile disease or other infectious diseases, such as diabetes and cardiovascular problems, showed similar rates between groups (Figure 4, Appendix A). Finally other infections, autoimmune disorders, pre-existing neurological problems, and smoking showed no differentiation between groups (Figure 4, Appendix A).

### 3.3. Treatment and Interventions

Treatment and other interventions were reported only in 70 cases of patients that recovered and 31 patients that succumbed. Treatment was mainly associated with the comorbidities, while general medications for encephalitis and meningitis included mainly combinations of antibiotics and antiherpetic drugs. Following diagnosis, specific treatments were used for WNV disease, such as intravenous immunoglobulin and in few cases interferon (Figure 5, Appendix A). The analysis highlighted a trend of increased usage of corticosteroids, anti-herpes drugs, and immunosuppressants, while plasmaphereses was more reported in the deceased group. However, these trends were of low statistical significance due to low reporting rate. The only intervention that varied with statistical significance was mechanical ventilation and intubation (~3.8 folds [recovered 4.29%/deceased 16.13%], *p* < 0.05) (Figure 5, Appendix A). The difference may be explained by the severity characteristics of the final stages of the disease in the deceased group. Assessment of a link between the occurrence of pulmonary disease and mechanical ventilation could now be confirmed (*p* = 0.6344, Appendix A).

## 4. Discussion

Outbreaks of West Nile virus disease occur annually around the globe from the Americas to Australia, reaching increasingly higher geographical latitudes. While such outbreaks are anticipated to increase in the decades to come due to climate change, urbanization, and travel [11], our understanding of WNV disease pathophysiology lags behind other infectious diseases. The understanding of the role of innate immunity and neuroinflammation in the recent years has given some insights into the understanding of the risk factors for neurological symptoms and fatal outcome. Previous reports on encephalitis incidence and death from WNV identified age, hypertension, chronic renal disease, infection with hepatitis C, immunosuppression, and diabetes as risk factors [12,13,14]. Such reports usually encompassed patients from single outbreaks or from series of outbreaks in the same region. In order to assess the effect of a variety of comorbidities and the predictive value of several symptoms on the negative outcome of WNV disease, we performed an extensive systematic comparison of published case series and case studies available on *NCBI PubMed* and retrieved through Rayyan platform.

Our study included 342 individuals that survived WNV infection and 97 individuals that did not. The patients in both groups presented any WNV-related disease entity that has been recorded. We focused on 32 symptoms, 17 comorbidities, 5 blood and CNS measurements, and 10 treatment and other clinical interventions that, in the past, had been described or used during diagnosis and clinical assessment of WNV disease severity. Although the presented final list of patients was not normalized, the groups were balanced in terms of sex ratio, age, and geographical origin (Appendix A).

According to our findings an altered mental status, delirium, confusion, fever, and anorexia were more prevalent amongst patients that succumbed (Figure 2, Appendix A). In the past, cognitive difficulties [15] and problems in concentration, memory, and understanding had been linked to more severe WNV disease reflected in a higher rate of hospitalization [16]. Notably, mental status problems are amongst the most prevalent long-term sequelae of WNV disease [7], especially in those patients with encephalitis symptoms. On the other hand, presentations such as headache, walking problems and poor balance, neck stiffness, and paresthesia were more prevalent amongst WNV survivors. Most of these symptoms are linked to meningitis that is well-documented in the past to be associated with less-severe forms of WNV disease [7]. However, as headache is a common symptom in either mild or severe disease, there is a possibility that it may be underreported in more severe cases where other neurological symptoms prevail. Rash was more prominent amongst the patients that recovered. This seems to be in agreement with the observation that rash is mostly associated with West Nile meningitis and WNF in contrast to the more severe West Nile encephalitis [7,17].

Measurements in CSF and blood regularly accompany molecular and serological diagnosis of WNV to assist in the assessment of viral meningoencephalitis [18]. In a study comparing 250 patients with West Nile virus meningitis and encephalitis, both groups presented similar degrees of CSF pleocytosis and normal CSF glucose, while patients with encephalitis had higher CSF protein concentrations [19]. Unfortunately, blood glucose was rarely reported, and thus, CSF/blood ratio could not be estimated for the majority of cases. In our study, survivors and non-survivors presented similar or non-statistically different levels of CSF protein and pleocytosis (Figure 3). Interestingly, while survivors presented normal levels of CSF glucose, non-survivors presented a statistically significant (*p* < 0.001) increase in CSF glucose (Figure 3, Appendix A). Taking into account that diabetes had similar prevalence between the two groups the importance of this finding requires further investigation.

A plethora of comorbidities have been associated in the past with more severe forms of WNV disease. As anticipated, conditions such as cancer and transplantation were highly associated with a poor outcome, which in the past has been attributed to an immunocompromised status of these patients. Innate and adaptive immune responses are key factors in both viral clearance and neuroinvasion, while immunosuppressants and age-related immunosenescence has been reported as precipitating factors of severe WNV disease [20]. An unanticipated result was the increased prevalence of pulmonary infections or other pulmonary diseases amongst patients that succumbed (Figure 4, Appendix A). Rare cases of severe neurological respiratory failure due to WNV have been reported in the past [21]. Diaphragmatic palsy or undiagnosed forms of neurological lung insufficiency may contribute to WNV mortality as exemplified by human cases [21,22] and mouse models [23]. Our results combined with previous evidence may support the requirement for pulmonary assessment of patients with WNV encephalitis. Other comorbidities that have been associated with WNV disease poor prognosis in previous studies are diabetes [24], hypertension, and cardiovascular disease [14,24,25,26]. In our study, all three conditions, although they presented a trend of increased incidence in non-survivors, did not show a statistically significant difference between groups (Figure 4, Appendix A). A meta-analysis and systematic review based on 18 individual studies reported that hypertension, diabetes, and heart diseases were the most prevalent in WNV patients and were linked to increased severity [27]. The mean age of the increased severity groups used in the above study and similar studies is significantly higher, and although the odds ratio of the results is frequently normalized, the effect of the aged subjects in the final outcome cannot be excluded. In our study, a 5-year difference between groups may accounted for the observed trend in the higher prevalence of diabetes, hypertension, and cardiovascular disease in the non-survivors group.

Regarding treatment, half of the cases were treated with antibiotics either alone or in combination with antiherpetic drugs against potential bacterial or herpes viral encephalitis and meningitis [28]. Use of corticosteroids or immunosuppressants to tackle inflammation or as treatment of other comorbidities did not have a significant difference between the two groups despite a trend towards increased usage in the group of patients that deceased. The value of corticosteroids despite some positive outcomes [29,30] is still under assessment [31]. On the other hand, immunosuppression especially amongst transplant recipients is considered to increase the risk of severe WND [32]. The most prominent difference between the recovered and deceased patient groups was the requirement for mechanical ventilation or intubation. In severe forms of WND, intubation and mechanical ventilation may be required because of a decreased level of consciousness, inability to clear secretions, or respiratory muscle weakness [33].

## 5. Conclusions

Our systematic review includes the largest collection of cases and case series on West Nile virus disease, focusing on the mortality determinants and revealing new insights in the clinical assessment of and effect of comorbidities on the disease outcome. While the pathophysiology of death following WNV disease is still mostly under-investigated, understudying the role of parameters such as CSF glucose and the effect of comorbidities such as pulmonary diseases is critical in the evaluation of death risk. Our study has several limitations mainly derived from the availability of published data. The certainty in our findings is limited by the fact that our review is based on case reports and case series. However, we included 428 reports that represent the best available published evidence to date. We pooled data from all patients and used simple statistical tests to explore differences between the survivors and non-survivors, and thus, this analysis is exploratory and at risk of publication bias, so we recommend a cautious interpretation of our findings. As the questions regarding susceptibility to lethal WNV disease remain open, there is a need for an international prospective registry to collect and evaluate WNV patient metadata. Furthermore, such a registry should also contain and compare metadata from other flaviviruses that can cause similar neurocognitive deficits and thus serve as a differential diagnosis platform for disease severity and death.

## Figures and Tables

**Figure 1 tropicalmed-07-00236-f001:**
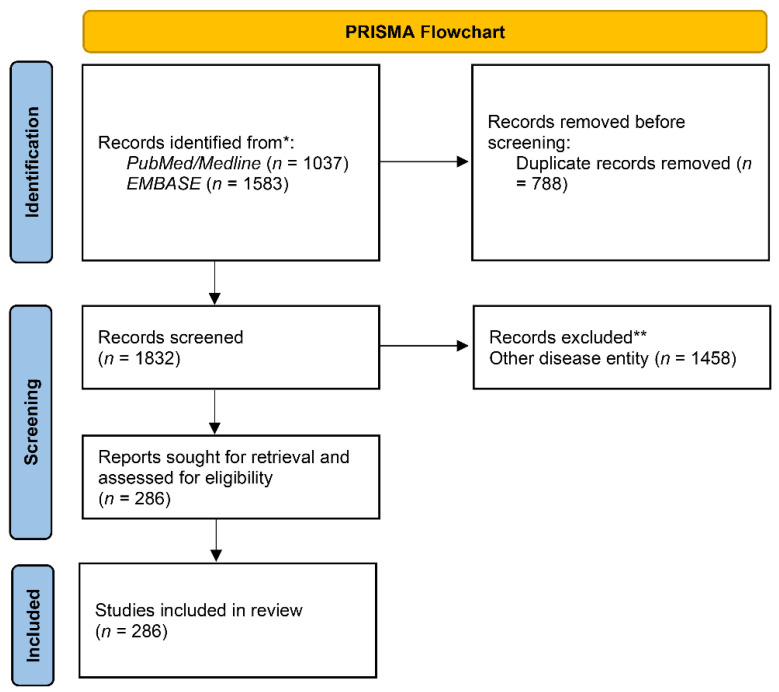
PRISMA 2020 flow diagram visually summarizes the screening process followed in the present report.

**Figure 2 tropicalmed-07-00236-f002:**
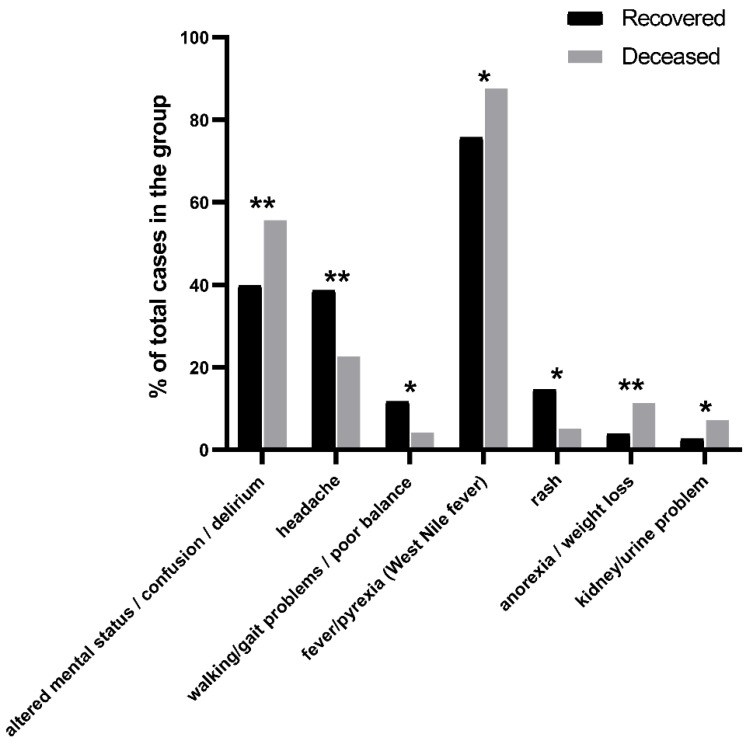
Rates of occurrence for the majority of symptoms related to WNV infection between patients that recovered and those that deceased. Significant differences for each symptom between the recovered and deceased groups of patients were assessed by a two-tailed *t*-test (two-sample assuming equal variances) and are indicated by (* *p* < 0.05 and ** *p* < 0.01).

**Figure 3 tropicalmed-07-00236-f003:**
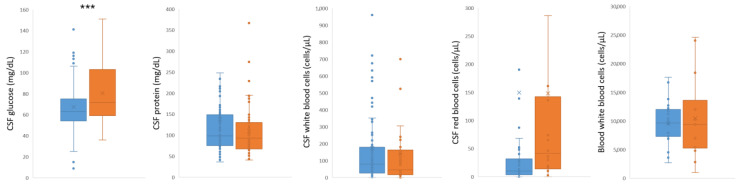
Average values of CSF and blood measurements between WNV-infected patients that recovered and those that deceased. Significant differences for measurement between the recovered and deceased groups of patients were assessed by a two-tailed *t*-test (two-sample assuming equal variances) and are indicated by (*** *p* < 0.001).

**Figure 4 tropicalmed-07-00236-f004:**
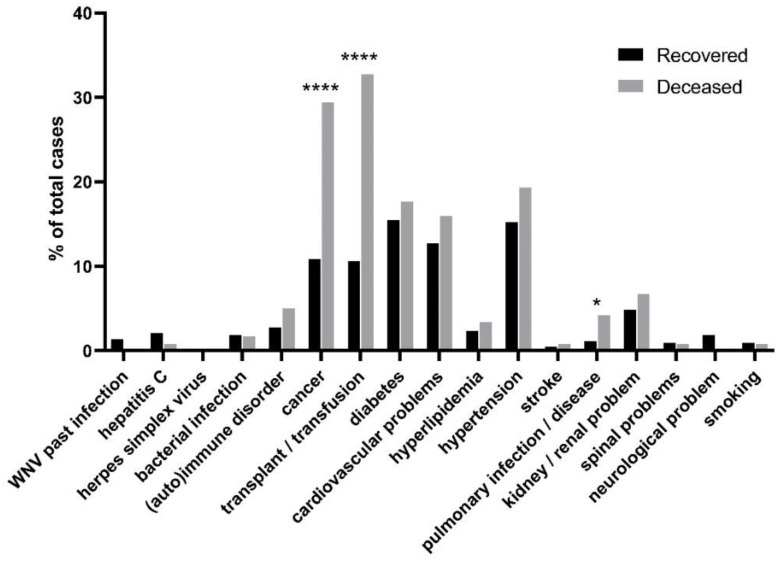
Rates of occurrence for the majority of comorbidities related to WNV infection between patients that recovered and those that deceased. Significant differences for each comorbidity between the recovered and deceased groups of patients were assessed by a two-tailed *t*-test (two-sample assuming equal variances) and are indicated by (* *p* < 0.05, and **** *p* < 0.0001).

**Figure 5 tropicalmed-07-00236-f005:**
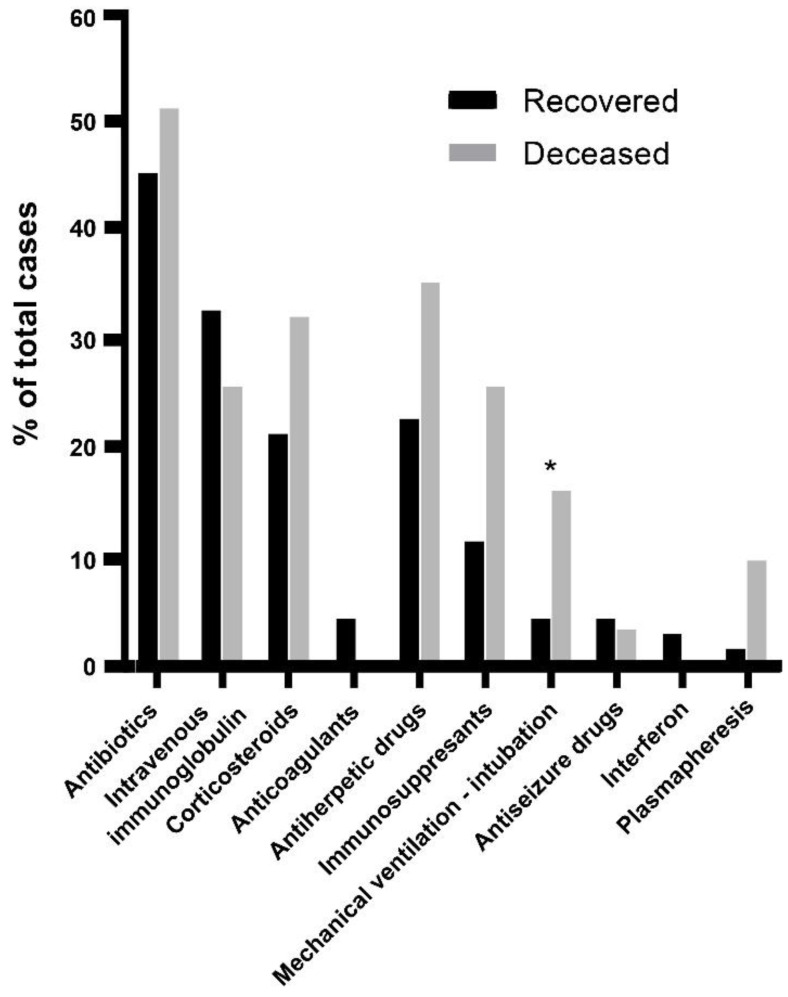
Rates of usage of various treatments and other clinical interventions following hospitalization between patients that recovered and those that deceased. Significant differences for each intervention between the recovered and deceased groups of patients were assessed by a two-tailed *t*-test (two-sample assuming equal variances) and are indicated by (* *p* < 0.05).

## Data Availability

All data used for this report are provided as Appendix A.

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
