# Peer review of "West Nile Disease Symptoms and Comorbidities: A Systematic Review and Analysis of Cases"

_tropicalmed, 2022, doi:10.3390/tropicalmed7090236_

Round 1

Reviewer 1 Report

Papers of this type are always interesting for researchers who deal with the mentioned problem and facilitate their understanding of the latest knowledge from it. Such works require knowledge of the subject matter and demonstrate the expertise of the researcher in the specified field. My opinion is that it should be published in your journal.

Author Response

We would like to thank the reviewer for his enthousiatic reception of our manuscript.

Reviewer 2 Report

Dear authors,

I would like to congratulate you on a very extensive systematic review on this interesting topic regarding WNV disease- clinical presentations and comorbidities. I have several concerns and suggestions which I hope will help improve the paper and hopefully result in publication. 

1. Abstract- summary of results should be provided in this section

2. Line 45- Powassan virus should be mentioned here as well

3. Introduction is very long- consider revising and making it shorter to improve the engagement of the reader and to help keep focus. Particularly in comparison to discussion ( which should be largest part of the paper) introduction is quite extensive. 

4. Lines 101-106- residual neurocognitive deficits are not specific only to WN neuro invasive disease. POWV encephalitis can cause similar sequela and parallel between the two should be elaborated in more details  in discussion section

5. Figure 2 is very difficult to follow, it was upside down and there are too many symptoms included in it. I would suggest to authors to cut symptoms in half and report only the ones where the difference between survivors and non survivors was the most evident

6. In my opinion all case reports that were included in analysis should be cited but I recognize that given the extent of cases analyzed this might be a lot of additional work so this is an optional suggestion

7. Discussion should also focus on similarities and differences between WN virus disease and other Flaviviridae particularly TB encephalitis and Powassan virus

8. One paragraph about treatment modalities  should be mandatory

9. Finally, the difference in treatment modalities, length of stay etc should be compared between survivors and non survivors

Author Response

We would like to thank the reviewer for his/her time to review our manuscript and for his/her contructive critisism over the following subjects.

C = Comment, R = Response

C1. Abstract- summary of results should be provided in this section.

R1. The summary of results has been added to the Abstract section.

C2. Line 45- Powassan virus should be mentioned here as well.

R2. Powassan virus and other important flaviviruses were added in the Introduction section.

C3. Introduction is very long- consider revising and making it shorter to improve the engagement of the reader and to help keep focus. Particularly in comparison to discussion ( which should be largest part of the paper) introduction is quite extensive.

R3. The manuscript has been changed accordingly.

C4. Lines 101-106- residual neurocognitive deficits are not specific only to WN neuro invasive disease. POWV encephalitis can cause similar sequela and parallel between the two should be elaborated in more details in discussion section.

R4. In the present report we did not perform differential analysis between diseases caused by different flaviviruses. This would be out of scope of our systematic review and we would open also a big Discussion section comparing WNV encephalitis and encephalitis cause by many other flaviviruses.

C5. Figure 2 is very difficult to follow, it was upside down and there are too many symptoms included in it. I would suggest to authors to cut symptoms in half and report only the ones where the difference between survivors and non survivors was the most evident.

R5. Figure 2 has been split in half for better reading. However, as all the symptoms included are often assessed or reported in clinical evaluation of WNV disease, we would like to retain the depiction of these details. In our first manuscript we had already removed the majority of rare symptoms and comorbidities.

C6. In my opinion all case reports that were included in analysis should be cited but I recognize that given the extent of cases analyzed this might be a lot of additional work so this is an optional suggestion.

R6. As citing 428 publications would flood the References section, we provide a separate table (Supplementary file 1) containing the aforementioned reports.

C7. Discussion should also focus on similarities and differences between WN virus disease and other Flaviviridae particularly TB encephalitis and Powassan virus.

R7. As stated in our answer for C4, the goal of our systematic review focused on the differences in pathology within the WNV patients. Thus, although we attempted, we could not embed a paragraph on differential diagnosis among flaviviruses. However, as we believe the point of view of our report is important for the deaths associated with other flaviviruses that can cause encephalitis, we propose in the Conclusions section that such a systematic review should also be conducted for other neurotropic viruses.

C8-C9. One paragraph about treatment modalities should be mandatory. Finally, the difference in treatment modalities, length of stay etc should be compared between survivors and non survivors.

R8-R9. We have added treatment and clinical interventions in the Results section. As very few reports included these parameters, the resulted bar chart could not be supported sufficiently by statistics and thus, we included the respective figure as supplementary. The difference in treatment modalities has been analyzed between recovered and deceased patients and the results of this analysis are now included in Supplementary Figure 2 and Supplementary Table 4.

Reviewer 3 Report

Dear Authors, below are a few comments/corrections I noted as I read through your manuscript:

Line 25: Delete "spread"

Line 56: Replace "It" with "WNF" (or spell it out since it is the start of a sentence - depending on journal formatting requirements

Line 72: Add a comma after "Usually"

Line 82: Delete "in" after "deceased"

Figure 2, 4, and 5: Y-axis - please define what the "group" is for the y-axis title or just have it as "% of total cases"

Section 3.1 and beyond: Please clarify if the "recovered" individuals you are referring to are recovered from WNND or any WNV-infection which would include WNF and WNND. 

Line 142: Move "often" to after "succumbed"

Section 3.2: It would be nice to add percentages and (x/y) for the numbers discussed 

Line 221: Add a comma after "past"

Line 262: Replace "noy" with "not"?

Line 272: I'm assuming you mean "herpes", not "hespes"?

Author Response

We would like to thank the reviewer for the comments

all line-specific changes have been made

Comment on Section 3.1 and beyond: Please clarify if the "recovered" individuals you are referring to are recovered from WNND or any WNV-infection which would include WNF and WNND. 

We have added a definition of the groups in the Materials and Methods and also in the respective point in the Discussion in order to make clear to the reader what is their disease background.

Comment on Section 3.2: It would be nice to add percentages and (x/y) for the numbers discussed 

We have added the percentages next to the fold differences in the section but also in other sections for uniformity.

Round 2

Reviewer 2 Report

Dear authors and the Editor,

I have read carefully your revision. I appreciate your thoughtful responses to my remarks. Unfortunately I remain hesitant to recommend acceptance based on the following , remaining issues:

1. Line 112 - the sentence does not make any sense, please re-write;

2. Figure 2 is still very confusing to me and the results of it have not been clearly discussed. For example, rash and paresthesia are combined and it seems that survivors were more frequently affected by these symptoms/signs. Rash is dermatological manifestation, paresthesia neurologic- why would they be combined? How do you explain your results? Furthermore, Figure 2 also depicts that survivors had more headaches- this should be elaborated to in discussion section;

3. Discussion remains relatively short and disproportionate to introduction. 

4. Line 177-182- I am unimpressed with your addition regarding the treatment. I think this part can be expended. For me as, as a clinician, it is important to understand from your paper what I can offer to the patients. Instead of insisting on having large and uninformative figure 2 , one figure or table with treatment variables and outcome would be much more useful

5. Abstract remains uninformative- why did not you list CSF glucose and more prominently describe pulmonary comorbidities as a risk factor for the disease? This is the main finding from your systematic review and it should be clear to reader immediately after reading the abstract

All in all, I do appreciate effort the authors invested but I am not pleased with the revisions provided. 

Author Response

Dear Editor

Dear Reviewers

thank you again for your time and effrort to evaluate our manuscript and to assist in a better presentation of our findings.

these are our answers 

Dear authors and the Editor,

I have read carefully your revision. I appreciate your thoughtful responses to my remarks. Unfortunately I remain hesitant to recommend acceptance based on the following , remaining issues:

  1. Line 112 - the sentence does not make any sense, please re-write;

The sentence was corrected

  1. Figure 2 is still very confusing to me and the results of it have not been clearly discussed. For example, rash and paresthesia are combined and it seems that survivors were more frequently affected by these symptoms/signs. Rash is dermatological manifestation, paresthesia neurologic- why would they be combined? How do you explain your results? Furthermore, Figure 2 also depicts that survivors had more headaches- this should be elaborated to in discussion section;

We reduced Figure 2 to the most significant findings and have corrected rash/paresthesia to rash as it was a typo. We elaborated more on the findings in the discussion section.

  1. Discussion remains relatively short and disproportionate to introduction. 

We discussed more our results and have significantly reduced introduction.

  1. Line 177-182- I am unimpressed with your addition regarding the treatment. I think this part can be expended. For me as, as a clinician, it is important to understand from your paper what I can offer to the patients. Instead of insisting on having large and uninformative figure 2 , one figure or table with treatment variables and outcome would be much more useful

We have dedicated a new section on treatment and have discussed it in the discussion section

  1. Abstract remains uninformative- why did not you list CSF glucose and more prominently describe pulmonary comorbidities as a risk factor for the disease? This is the main finding from your systematic review and it should be clear to reader immediately after reading the abstract

We have elaborated more on these to findings but due to the 200-word restriction we could only add one more sentence on their importance as death risk factors.

All in all, I do appreciate effort the authors invested but I am not pleased with the revisions provided. 

We you like to thank you again for your constructive criticism.